# MusicAOG: an Energy-Based Model for Learning and Sampling a Hierarchical Representation of Symbolic Music

## Abstract

In addressing the challenge of interpretability and generalizability of artificial music intelligence, this paper introduces a novel symbolic representation that amalgamates both explicit and implicit musical information across diverse traditions and granularities. Utilizing a hierarchical and-or graph representation, the model employs nodes and edges to encapsulate a broad spectrum of musical elements, including structures, textures, rhythms, and harmonies. This hierarchical approach expands the representability across various scales of music. This representation serves as the foundation for an energy-based model, uniquely tailored to learn musical concepts through a flexible algorithm framework relying on the minimax entropy principle. Utilizing an adapted Metropolis-Hastings sampling technique, the model enables fine-grained control over music generation. A comprehensive empirical evaluation, contrasting this novel approach with existing methodologies, manifests considerable advancements in interpretability and controllability. This study marks a substantial contribution to the fields of music analysis, composition, and computational musicology.

## 1 Introduction

### 1.1 Motivation

Music generation methodologies can traditionally be segmented into two primary representations: symbolic and audio (Ji et al., 2023). Models within the audio domain, such as MusicLM Agostinelli et al. (2023), generate direct audio output. Notably, achieving control in audio generation primarily relies on natural language techniques. However, encapsulating musical nuances within natural language descriptors remains an intricate task (Kim & Belkin, 2002)(Davies, 1983)(Clarke, 1989). A deeper issue emerges when considering the inherent difficulty in intuitively understanding frequency spectrums. Hence, symbolic representations emerge as a promising alternative, providing a sparse coding interface that bridges the chasm between human intuition and the high-entropy nature of digital audio samples, especially in tasks requiring detailed musical control.

Deep learning paradigms tailored for symbolic music generation often specialize in specific facets, be it conditional generation, inpainting, or the creation of melody, harmony, and accompaniment structures (Ji et al., 2023). These models, although intricate, often lack a holistic theoretical framework and face hurdles in extrapolating to more comprehensive musical composition tasks. Another challenge is that many of these models adapt loss functions originally devised for natural language processing or computer vision. Such borrowings, while innovative, might not encapsulate musical intricacies fully, hampering control over output generation.

A few symbolic music generation models, like Hyun et al. (2022), Zou et al. (2021), and Wang et al. (2020), have embarked on integrating hierarchical and modifiable structures. Although these models incorporate a degree of interpretability and control, their granularity is often not meticulous enough to encompass the breadth of prior knowledge in symbolic music.

While the Generative Theory of Tonal Music (GTTM) by Lerdahl & Jackendoff (1983) presents an insightful hierarchical blueprint for musical analysis, current adaptations of GTTM predominantly

cater to homophonic tonal melodies, leaving out compositions enriched with polyphonic non-tonal components.

In light of these identified limitations, our research endeavors to pioneer methods that surmount challenges in both control and generalization within symbolic music generation. We envisage an approach interweaving hierarchical constructs with visually intuitive representations, aiming for versatile control over the generation mechanism and broad applicability across diverse musical traditions.

## 1.2 CONTRIBUTION

In summary, this paper makes three contributions:

- We propose a flexible and generalized hierarchical representation for symbolic music, encompassing existing representations while also providing additional musicological insights.
- The developed energy-based model, based on this hierarchical representation, enables effective learning of score music with interpretability.
- By adapting and employing Metropolis-Hastings sampling, we enable controlled generation of music, granting users fine-grained control over compositions.

## 2 RELATED WORK

The **stochastic and-or grammar** (AOG) initially emerged for parsing visual images(Zhu & Mumford, 2006). It later extended to temporal and causal patterns in videos and robotic activities(Xiong et al., 2016)(Pei et al., 2011)(Shu et al., 2015). The attributed and-or graph (A-AOG) was introduced to enhance attribute reasoning (Park et al., 2016), and has been beneficial in scene synthesis(Qi et al., 2018). We adapt the A-AOG representation and synthesis methodologies to symbolic music in our research.

The **filters random fields and maximum entropy (FRAME) model**, rooted in the minimax entropy principle, was crafted for texture characterization(Zhu et al., 1997). Recognized as energy-based models, they've been utilized for analyzing natural images, unveiling detailed texture properties(Lu et al., 2016a). The FRAME model was later applied to data articulated through and-or graphs(Zhu & Mumford, 2006). In our work, we refactor the FRAME model to tailor the learning mechanism for music, aiding in interpreting musical nuances.

The **Metropolis-Hastings algorithm**, a key part of the Markov chain Monte Carlo (MCMC) sampling techniques(Metropolis et al., 2004)(Hastings, 1970), has been effective in tasks like furniture arrangement and indoor scene synthesis(Yu et al., 2011)(Qi et al., 2018). We adapt this algorithm to enhance music generation while allowing fine control over the modification of existing musical constructs.

**The Generative Theory of Tonal Music (GTTM)** provides a robust framework for understanding music perception and cognition based on generative grammar principles(Lerdahl & Jackendoff, 1983). We extract the core of GTTM, merging its four main concepts into a unified entity. During implementation, we included specific well-formedness rules to narrow down the search domain during the sampling process.

## 3 REPRESENTATION OF SYMBOLIC MUSIC

This study utilizes the attributed And-Or Graph (A-AOG) as the representation of symbolic music. The A-AOG is a stochastic hierarchical grammar model that integrates phrase structure grammar, dependency grammar, and attribute grammar. The specific A-AOG used for music, known as **MusicAOG**, can be described by a 6-tuple:

$$\mathcal{AG}_{\mathrm{mus}} = <S, V, E, \mathcal{R}, X, \mathcal{P}> \tag{1}$$

We will now provide a breakdown of each component:

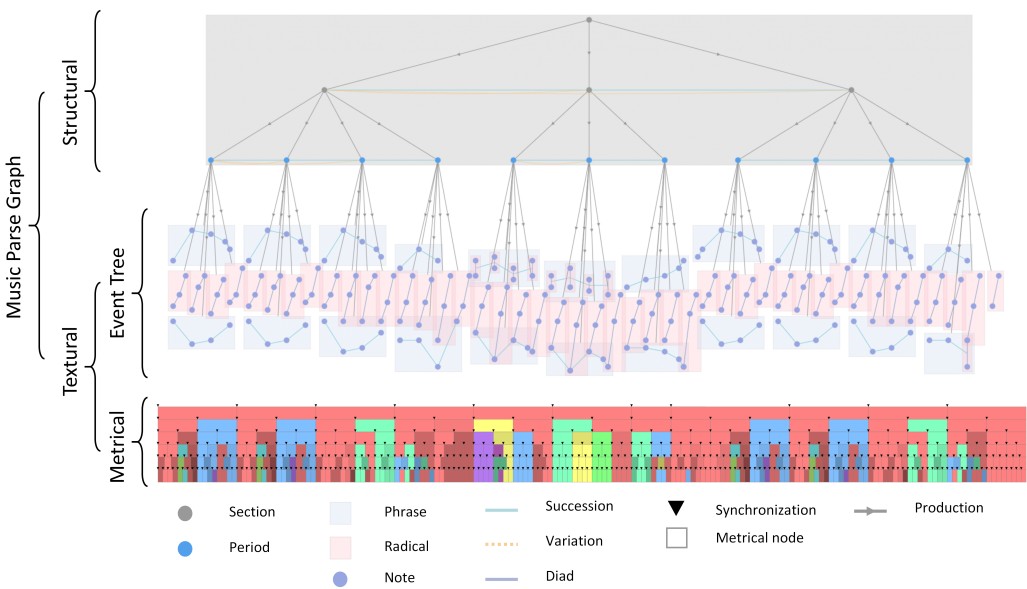

Figure 1: An example music parse graph of Robert Schumann's Kinderszenen, Op.15, No.1: Von fremden Ländern und Menschen. The section, period, phrase, radical, note and metrical node are nodes on the parse graph, while the succession, variation, diad, and synchronization are relations. They are arranged hirarchically and horizontally throu production rules and relations. The metrial tree are implicit tree where each metrical nodes are linked to nodes in event trees with synthronization relations.

1) $S$ represents the root node for the concept of music.

2) The vertex set $V = V_{\mathrm{and}} \cup V_{\mathrm{or}} \cup V_{\mathrm{T}}$ consists of three distinct subsets: (i) $V_{\mathrm{and}}$ represents a set of and-nodes responsible for decomposing musical ideas; (ii) $V_{\mathrm{or}}$ represents a set of or-nodes that allow branching to alternative decompositions and facilitate reconfiguration; (iii) $V_{\mathrm{T}}$ comprises a set of terminal nodes that ground the representation in radicals. Each node $v \in V$ is associated with state variables indicating the temporal onset and offset of the musical event $(t_{\mathrm{on}}, t_{\mathrm{off}})$.

3) The edge set $E = E_{\mathrm{succ}} \cup E_{\mathrm{var}} \cup E_{\mathrm{diad}} \cup E_{\mathrm{syn}}$ consists of four distinct subsets: (i) $E_{\mathrm{succ}}$ represents a set of edges denoting the succession of musical events; (ii) $E_{\mathrm{var}}$ represents a set of edges enabling repetition, variation, and recapitulation of musical ideas; (iii) $E_{\mathrm{diad}}$ captures generalized harmonic intervals between pitched musical events; (iv) $E_{\mathrm{syn}}$ ensures the synchronization of rhythmic patterns.

4) $\mathcal{R} = \{\gamma_1, \ldots, \gamma_k\}$ stands for the set of production rules. It describes how nodes in the AOG are decomposed into child nodes, representing the hierarchy of the AOG.

5) The attribute set $X = \{\mathbf{x}_1, \ldots, \mathbf{x}_k\}$ is associated with the nodes in $V$. The attributes are defined based on the and-node and can include additional dimensions for further learning tasks. The "pitch" attribute is represented using a vector instead of a scalar MIDI pitch. Temporal attributes $(ton, toff)$ are treated separately. Attributes can be propagated between parent and child nodes in the hierarchy.

6) $\mathcal{P}$ represents the probability model associated with the graphical representation.

A MusicAOG serves as a conceptual framework for music, encompassing the entire music grammar and containing all valid music. A parse graph is an instance generated by the AOG, representing a music piece through the switching of children of or-nodes. Formally, a parse graph can be defined as:

$$\mathbf{pg} =< V, E, \mathcal{R}, X > . \tag{2}$$

In the above notation, $E$ represents the set of edges, $\mathcal{R}$ represents the production rules, and $X$ represents the set of vertices. The set of nodes $V$ comprises only and-nodes and terminal nodes, i.e., $V = V_{\text{and}} \cup V_{\text{T}}$.

In an and-or grammar, concepts are represented through or-nodes, which are associated with instances of those concepts represented by child and-nodes. In natural language and vision domains, the number of or-nodes is typically large, approximately equivalent to the number of words in a natural language dictionary. However, defining concepts in the context of music composition is often ambiguous. Therefore, in MusicAOG, we limit the representability of or-nodes to capture explicit or implicit "fragments" of music and refer to nodes with different possible attributes as different **types** of nodes, compared to different vocabulary items in an and-or grammar. Each or-node has a single child and-node, and the switching of an or-node corresponds to the production of attributes for that particular and-node:

$$A \rightarrow (\beta, \mathbf{x}), \text{with } A \in V_{\text{or}}, \ \beta \in (V_{\text{and}} \cup V_{\text{T}})^+, \ \mathbf{x} \in X. \tag{3}$$

Subsequently, the attributed and-node generates one or several or-nodes along with their associated relations $r$ (together known as the configuration) through a production rule $\gamma$. The specific number of child or-nodes depends on the attributes of the parent and-node $n(\mathbf{x})$. Figure 1 provides a intuitive visualisation of a parse graph [1], and the details are described as below.

## 3.1 NODES

Specifically, nodes in MusicAOG can be **textural** and **structural**, i.e. $V = V_{\text{text}} \cup V_{\text{struct}}$. the structural nodes at the structural level encapsulate **section** nodes and **period** nodes which represent temporal segments of music, i.e. $V_{\text{struct}} = V_{\text{section}} \cup V_{\text{period}}$. Period nodes represent short timescales, focusing on distinct cognitive processes like emotions, styles. It is produced by section nodes of longer timescale, emphasizing more reflective elements such as narratives behind the music.

At the textural level, period nodes are further decomposed into **phrases** and **radicals** within an event tree, while the rhythmic and harmonic aspect of the music is captured by the metrical tree. This layered representation aids in capturing both the explicit and implicit elements of musical composition "spatially", contributing to a richer understanding and analysis of musical pieces. Detailed explanation of these nodes can be found in the Supplementary Information.

## 3.2 RELATIONS

In the realm of musical composition, relations or edges are pivotal as they instill a defined structure, akin to a dependency grammar in language, orchestrating how musical elements interact and unfold over time. Among these relations, **succession** relations are prominent, painting a picture of how musical fragments succeed one another. This succession, dictated by various variables, showcases not just the transition but the temporal distance, overlapping, and the smoothness of how one musical fragment flows into the next.

On another hand, **diad** relations elucidate the vertical pitch intervals between radicals, while the **synchronization** relations ($E_{\text{syn}}$) act as pointers that bridge metrical nodes and radicals, specifying which musical time interval a node belongs to.

In many musical compositions, the recapitulation of musical elements enhances memorability. This is represented by the **variation** relations ($E_{\text{var}}$). It is important to note that, unlike traditional music theory, variation relations can encompass not only variations but also repetitions and sequences. The variation relations also possess attributes, although the details are beyond the scope of this paper. In the Supplementary Information, we propose simple formulation of these relations.

---

[1] https://musescore.com/user/23490456/scores/5367341

## 4 PROBABILISTIC FORMULATION

In this study, the MusicAOG is represented using a descriptive model. Utilizing the maximum entropy principle, the prior probability of a given music parse graph can be expressed according to the Gibbs distribution as:

$$p(\mathbf{pg}; \Theta, E, \Delta) = \frac{1}{Z(\Theta)} \exp\left(-\mathcal{E}(\mathbf{pg}; \Theta, E, \Delta)\right) \tag{4}$$

Here, the energy term $\mathcal{E}(\mathbf{pg}; \Theta, E, \Delta)$ is the sum of all energy components associated with the attributes of nodes, relations, and the production of or-nodes that are derived from the AOG:

$$\mathcal{E}(\mathbf{pg}; \Theta, E, \Delta) = \mathcal{E}_\Theta(X(V_{\text{and}})) + \mathcal{E}_\Theta(E) + \mathcal{E}_\Theta(V_{\text{or}}) \tag{5}$$

In the domain of music, the aforementioned terms can be dissected into constituent parts involving structural and textural nodes, along with four distinct types of relations:

$$\begin{aligned}
\mathcal{E}(\mathbf{pg}; \Theta, E, \Delta) =& \mathcal{E}_\Theta(X(V_{\text{struct}})) + \mathcal{E}_\Theta(X(V_{\text{text}})) + \mathcal{E}_\Theta(E) + \mathcal{E}_\Theta(V_{\text{or}}) \\
=& \mathcal{E}_\Theta(X(V_{\text{section}})) + \mathcal{E}_\Theta(X(V_{\text{period}})) + \mathcal{E}_\Theta(X(V_{\text{phrase}})) + \mathcal{E}_\Theta(X(V_{\text{radical}})) \\
& + \mathcal{E}_\Theta(E_{\text{succ}}) + \mathcal{E}_\Theta(E_{\text{var}}) + \mathcal{E}_\Theta(E_{\text{diad}}) + \mathcal{E}_\Theta(E_{\text{sync}}) + \mathcal{E}_\Theta(V_{\text{or}})
\end{aligned} \tag{6}$$

As delineated in a preceding section, the MusicAOG permits any or-node to generate a substantial set of configurations from possible combinations of child nodes. Consequently, enumerating all possible configurations as a multinomial distribution is infeasible. Instead, the approach taken is to evaluate the energies of the edges connecting parent nodes to the child nodes they produce (the production rules $\mathcal{R}$). A multinomial distribution is employed solely to characterize the number of child nodes associated with a parent node, denoted as $V_{\text{num}}^{\text{or}}$:

$$\mathcal{E}_\Theta(V_{\text{or}}) = \mathcal{E}_\Theta(\mathcal{R}) + \mathcal{E}_\Theta(V_{\text{num}}^{\text{or}}) \tag{7}$$

## 5 LEARNING MUSICAOG

The aim in learning MusicAOG is to minimize the Kullback-Leibler divergence between the probabilistic model $p(\mathbf{pg})$ and the true distribution of symbolic music $f$. Given a sufficiently large sample size $N$, this process is equivalent to a maximum likelihood estimation (MLE):

$$p^* = \underset{p \in \Omega_p}{\arg\min} D_{\text{KL}}(f\|p) \approx \underset{p \in \Omega_p}{\arg\max} \frac{1}{N} \sum_{i=1}^{N} \log p(\mathbf{pg}_i; \Theta, E, \Delta) \tag{8}$$

Under the descriptive method, a set of feature statistics $\phi_\alpha(\mathbf{pg}), \alpha = 1, \ldots, K$ is defined. These feature statistics provide the necessary constraints for the formulation of the model $p$. Formally, this is represented as:

$$\Omega_p = \{p(\mathbf{pg}) : \mathbb{E}_{p(\mathbf{pg};\Theta)}[\phi_\alpha(\mathbf{pg})] = \mathbf{h}_\alpha, \alpha = 1, \ldots, K\} \tag{9}$$

Here, $\mathbb{E}[\phi_\alpha(\mathbf{pg})]$ represents the marginal distribution of $p$ with respect to observed statistics, and $\mathbf{h}_\alpha$ denotes the histogram from the application of the $\alpha$-th feature on the parse graph.

### 5.1 MAXIMUM ENTROPY FOR PARAMETER LEARNING

Define the log-likelihood as:

$$\mathcal{L}(\mathbf{pg}; \Theta) = \frac{1}{N} \sum_{i=1}^{N} \log p(\mathbf{pg}i; \Theta). \tag{10}$$

Through the application of Lagrange multipliers for MLE under the maximum entropy principle and setting $\frac{\partial \mathcal{L}(\mathbf{pg};\Theta)}{\partial \lambda} = 0$, we obtain the exponential form:

$$p(\mathbf{pg}; \Theta, E, \Delta) = \frac{1}{Z(\Theta)} \exp\left(-\sum_{\alpha=1}^{K} <\lambda_\alpha, \mathbf{h}_\alpha>\right), \text{where } \Theta = \lambda_\alpha, \alpha = 1, \ldots, K \tag{11}$$

---

**Algorithm 1:** Learning MusicAOG via Minimax Principle

---

**Input:** Dataset of music parse graphs $D_{\text{obs}} = \{\mathbf{pg}_i^{\text{obs}}, i = 1, \ldots, N\}$, statistical feature bank
      $\mathcal{B} = \{\mathcal{F}_\alpha, \alpha = 1, \ldots, K\}$, learning rate $\eta$, total learning iterations $L$, error tolerance
      $\epsilon = 0.1$, sampling iterations $s$

**Initialize:** Selected feature bank $\mathcal{S} = \emptyset$, parameter set $\{\lambda_\alpha\} = \mathbf{0} \ \forall \ \alpha = 1, \ldots, K$, initial
      synthesized music parse graphs $D_{\text{syn}} = \{\mathbf{pg}_i^{\text{syn}} | i = 1, \ldots, N\}$ utilizing the
      sampling initialization procedure from subsection 6.1

**repeat**

    **for** $\mathcal{F}_{\alpha'} \in \mathcal{B} \backslash \mathcal{S}$ **do**

        |  Compute histograms $\mathbf{h}_{\alpha'}^{\text{obs}}$ and $\mathbf{h}_{\alpha'}^{\text{syn}}$;

    **end**

    Utilizing $\mathbf{h}_{\alpha'}^{\text{obs}}$ and $\mathbf{h}_{\alpha'}^{\text{syn}}$, select the optimal feature $\mathcal{F}_+^*$ from set $\{F_{\alpha'}\}$ following
     Equation 12;

    Update the selected feature set $\mathcal{S} \leftarrow \mathcal{S} \cup \{\mathcal{F}_+^*\}$;

    Set iteration counter $l = 0$;

    **repeat**

        Calculate $\mathbf{h}_+^{\text{obs}}$ and $\mathbf{h}_+^{\text{syn}}$ for currently selected features, use $\mathbf{h}_+^{\text{syn}}$ as $\mathbb{E}_{p(\mathbf{pg};\Theta)}[\phi_+(\mathbf{pg})]$;

        Update $\lambda_+$ through gradient descent: $\delta\lambda_+ = \eta(\mathbf{h}_+^{\text{syn}} - \mathbf{h}_+^{\text{obs}})$;

        Generate a new set of parse graphs $D'_{\text{syn}}$ from the existing $D_{\text{syn}}$ employing the method
         described in subsection 6.1 over $s$ iterations, subsequently updating the histograms;

        Increment iteration counter: $l \leftarrow l + 1$;

    **until** $l = L$ *or* $\mathbf{h}_+^{\text{syn}} - \mathbf{h}_+^{\text{obs}} < \epsilon$;

**until** *All statistical features in the feature bank are selected*;

---

This is consistent with the Gibbs form described in Equation 4, where the energy term $\mathcal{E}(\mathbf{pg}; \Theta, E, \Delta)$ corresponds to the parameterized potential function $\sum_{\alpha=1}^{K} \langle \lambda_\alpha, \mathbf{h}_\alpha \rangle$. Analogous to the additive decomposition of the energy term in prior sections, parameters $\lambda_\alpha$ can be estimated by extracting features of MusicAOG with following steps:

1. **Node Attributes Learning:** We learn the potential function $\lambda_{x(u)}$ for attributes on and-nodes and terminal nodes $u \in V_{\text{and}} \cup V_{\text{T}}$. Setting $\frac{\partial \mathcal{L}(\mathbf{pg};\Theta)}{\partial \lambda} = 0$, we derive the statistical constraints as $\mathbb{E}_{p(\mathbf{pg};\Theta)}[\phi_\alpha^{\text{node}}(x(u))] = \mathbf{h}_{x(u)}^{\text{obs}}$. This formulation is crucial for empirical distribution in subsequent data sampling.

2. **Relation Learning:** For relations $(s, t) \in E$, the potential function $\lambda_{s,t}$ is learned. Using the previously mentioned condition for maximization, we determine the statistical constraints $\mathbb{E}_{p(\mathbf{pg};\Theta)}[\phi_\alpha^{\text{relation}}(s, t)] = \mathbf{h}_{s,t}^{\text{obs}}$.

3. **Production Learning:** For relations $(m, n) \in \mathcal{R}$, the potential function $\lambda_{m,n}$ is determined. The maximization condition yields the statistical constraints $\mathbb{E}_{p(\mathbf{pg};\Theta)}[\phi_\alpha^{\text{prod}}(m, n)] = \mathbf{h}_{m,n}^{\text{obs}}$.

4. **Or-node Learning:** The MLE for the selection in or-nodes $\lambda_v, \forall v \in V_{\text{or}}$ is equivalent to the frequency of selected child counts of and-nodes by or-nodes. This is mathematically represented as $\mathbf{h}_v^{\text{obs}} = \frac{\#(n(v)=j)}{\sum_{j=1}^{n(v)} \#(n(v)=j)}, j = 1, \ldots, n(v)$.

Given that $\mathbb{E}[\phi_\alpha(\mathbf{pg})]$ is inaccessible directly, we employ Metropolis-Hastings sampling to generate a set $\{\mathbf{pg}_i^{\text{syn}}, i = 1, \ldots, N\}$. The synthesized histogram $\mathbf{h}_\alpha^{\text{syn}}$ is computed for $\mathbb{E}[\phi_\alpha(\mathbf{pg})]$. Details on the sampling process are provided in the succeeding section.

## 5.2   Minimum Entropy for Feature Selection

For the selection of salient statistical features (descriptors) from a feature bank $\mathcal{B} = \{\mathcal{F}_\alpha, \alpha = 1, \ldots, K\}$, we adhere to the minimum entropy principle. The criterion is that the chosen feature should most significantly reduce $D_{\text{KL}}(f \| p(\mathbf{pg}_i; \Theta, E, \Delta))$. This objective is realized by selecting

a feature that amplifies the disparity between observed and synthesized histograms:

$$\mathcal{F}_+^* = \underset{\mathcal{F}_+ \in \mathcal{B}}{\arg\max}\, D_{\mathrm{KL}}(f\|p) - D_{\mathrm{KL}}(f\|p_+) = \underset{\mathcal{F}_+ \in \mathcal{B}}{\arg\max}\|\mathbf{h}_+^{\mathrm{obs}} - \mathbf{h}_+^{\mathrm{syn}}\| \tag{12}$$

When integrating the maximum and minimum entropy principles, the comprehensive learning algorithm is elucidated in algorithm 1.

# 6  SAMPLING SCHEME

Music generation is realized by sampling a **pg** from the prior distribution characterized by the MusicAOG model. Direct sampling of node attributes and the number of child branches for or-nodes is facilitated using $\mathbf{h}_{x(u)}^{\mathrm{obs}}$ and $\mathbf{h}_v^{\mathrm{obs}}$. However, to sample relations and hierarchical structures that adhere to multiple joint constraints from the learned MusicAOG is non-trivial. To address this, we employ the Metropolis-Hastings algorithm within a Markov Chain Monte Carlo (MCMC) framework to sample music parse graphs.

## 6.1  DIRECT SAMPLING

The sampling methodology constructs music parse graphs following a top-down and sequential approach. Commencing from the root, for any non-terminal nodes, node types, attributes, and child counts are proposed based on their prior distribution. Once all children of a particular node are sampled, a random selection of two distinct child nodes is made, establishing a variation relation $E_{\mathrm{var}}$ between them.

Subsequent to the inital construction of the parse graph, several proposal mechanisms are available for the Hastings-Metropolis algorithm:(1) Propose a new attribute vector for a selected node in the parse graph, based on its prior distribution. (2)Propose the addition or removal of a child from a non-terminal node, or a variation relation $E_{\mathrm{var}}$.

Given a proposal, its acceptance is governed by the probability:

$$\alpha(\mathbf{pg}'|\mathbf{pg};\Theta) = \min\left(1, \exp\left(\mathcal{E}(\mathbf{pg};\Theta) - \mathcal{E}(\mathbf{pg}';\Theta)\right)\right). \tag{13}$$

It should be noted that the relations of succession, diad, and synchronization are automatically deduced post-proposal. While they are inherently used to evaluate the energy function $\mathcal{E}(\mathbf{pg}';\Theta)$, they also play a role in determining the proposal's acceptance probability.

## 6.2  CONTROLLED AMENDMENT

In the controlled amendment phase, each node attribute within a music parse graph is associated with a regulatory term, $T \geq 0$ (termed as temperature), which the user sets to indicate the desired amendment degree. During the continuation of MCMC sampling via Metropolis-Hastings on the parse graph, proposals are generated randomly but only pertain to attributes with $T > 0$. Proposals related to adding or deleting nodes and relations are valid when the parent node's temperature is greater than 0. The proposal's acceptance probability is subsequently adjusted by the temperature:

$$\alpha(\mathbf{pg}'|\mathbf{pg};\Theta) = \min\left(1, \exp\frac{(\mathcal{E}(\mathbf{pg};\Theta) - \mathcal{E}(\mathbf{pg}';\Theta))}{T_t(x')}\right). \tag{14}$$

The temperature term undergoes adjustments over iterations, adopting a simulated annealing approach, where $T_t = \frac{T_0}{\ln(1+t)}$.

To provide intuition: for $0 < T < 1$, modifications to the target property are subtle, aligning closely with the original music. When $T > 1$, the property is actively altered, even if it conforms to a plausible distribution, paving the way for exploring a broader creative musical space.

# 7  EXPERIMENT AND DISCUSSION

The representation advocated in this work is extensive and encompasses numerous elements. Furthermore, the learning algorithm is significantly dependent on thorough labeling carried out by professional musician, who engage in the study and construction of music parse graphs. To empirically

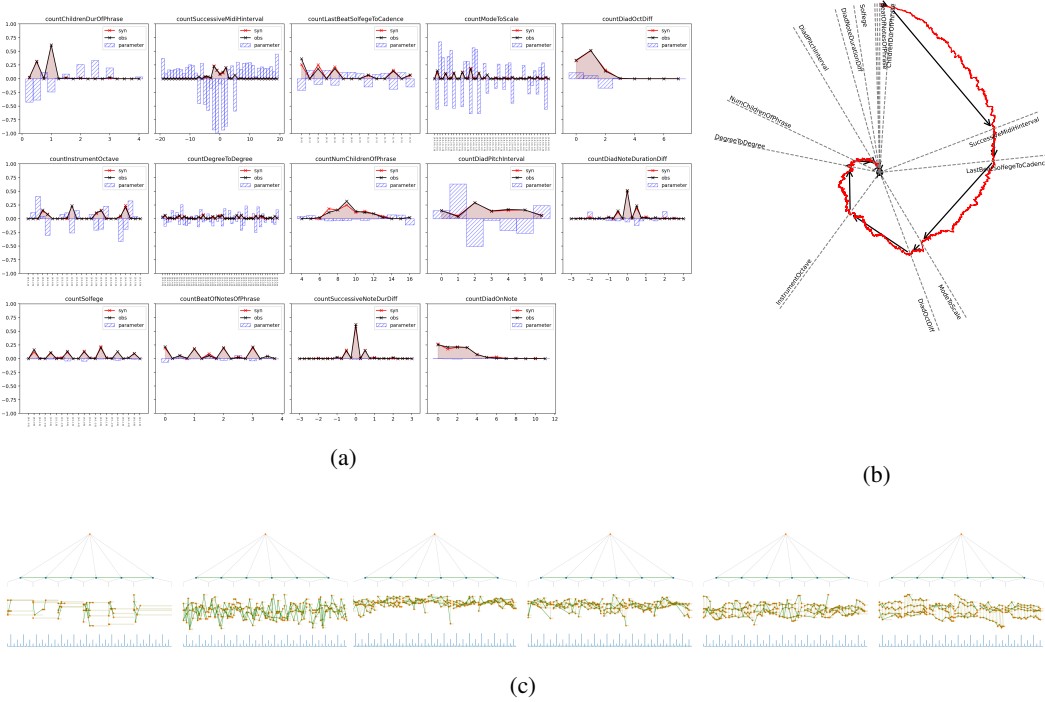

(a)

(b)

(c)

Figure 2: (a) The histograms of each features applied.(b) Information projections. (c) Sample from noise

ascertain the effectiveness of MusicAOG, we streamlined our model while preserving the indispensable concepts in AOG representation. A controlled experiment was undertaken utilizing 3 pieces from Bach's chorale (BWV 9, BWV 347, and BWV 267) sourced from MuseScore[2]. The **pg** for the chosen music score was meticulously constructed by an expert musician acquainted with our representational framework. The simplifications enacted during MusicAOG construction included: treating individual notes as radicals, disregarding variation relations, and simplifying the complex structures of metrical trees and synchronization relations to just three levels—measure-level beats giving rise to crotchet-level beats, which further devolve into semiquaver-level beats. At the textural level of the MusicAOG, a homogeneous distribution for nodes and edges is assumed. Consequently, the devised feature bank solely accentuates statistical note features. To ascertain robustness, even within this streamlined experiment, fourteen feature descriptors were conceived to encapsulate node attributes, relations, productions, and or-node selections (see histograms in Figure 2c).

The learning algorithm, delineated in algorithm 1, was configured with the following hyperparameters: maximum learning iterations per selected feature, $L = 500$; error tolerance $\epsilon = 0.1$; sampling iterations, $s = 150$; and learning rate, $\eta = 1$. For the controlled resampling test, the original parse graph data was employed. All notes' attributes were assigned a temperature of $T = 0.1$. Over 100 resampling iterations were executed to amend the original music composition. To mitigate complexity, proposals during sampling were solely node-centric, encompassing attribute alterations (adjusting pitches) and node insertions and deletions (adjusting durations and rhythms).

Figure 2b displays the evolution of energy values across learning iterations, in tandem with the sequence of feature selection. This plot intuitively illustrates how the minimax entropy principle in the learning algorithm projects the model distribution onto the eligible feature defined by $H_k$ for $k$-th feature selection, and gradually approximates the true distribution. In the realm of controlled generation, the notes originating from noise $\mathbf{pg}_{\text{syn}}^0$ undergo resampling to gradually yield $\mathbf{pg}_{\text{syn}}^*$ under addition of descriptors showcased in Figure 2b. A comparative analysis between the histograms corresponding to the observed $\mathbf{pg}^{\text{obs}}$ and the synthesized $\mathbf{pg}^{\text{syn}}$ for each feature descriptor is provided in Figure 2c, alongside parameters on each bin. These findings suggest that despite the limited parameter space (only 289 parameters), the model has adeptly assimilated all the features,

---

[2]https://musescore.com/user/11015626/scores/3117011

Table 1: Comparison of subjective scoring of music generated by different model

| Methods | Style Coherence | Structural Clarity | Integrity | Balance of Voice | Textural Rationality | Notation Professionality | Playability |
|---|---|---|---|---|---|---|---|
| Lv et al. (2023) | $7.82 \pm 1.90$ | $5.73 \pm 1.86$ | $6.91 \pm 2.02$ | $7.00 \pm 1.86$ | $7.36 \pm 2.06$ | $7.64 \pm 1.55$ | $7.82 \pm 1.11$ |
| Hyun et al. (2022) | $8.45 \pm 1.30$ | $6.00 \pm 1.60$ | $4.73 \pm 1.29$ | $6.18 \pm 1.03$ | $7.73 \pm 1.60$ | $7.64 \pm 1.82$ | $8.55 \pm 0.99$ |
| Lu et al. (2023) | $7.91 \pm 1.78$ | $6.27 \pm 1.21$ | $4.91 \pm 1.50$ | $7.27 \pm 1.66$ | $7.36 \pm 1.97$ | $4.36 \pm 1.67$ | $7.55 \pm 1.88$ |
| MusicAOG (ours) | $8.91 \pm 1.16$ | $8.09 \pm 1.62$ | $8.09 \pm 2.19$ | $8.73 \pm 1.29$ | $8.73 \pm 1.42$ | $9.18 \pm 0.94$ | $9.27 \pm 0.86$ |

Table 2: Comparison of controlling condition of different approaches. The size of dataset used for training and the size of parameters of model are also compared.

| Methods | Control Condition | Size of Dataset | Size of Parameters |
|---|---|---|---|
| Lv et al. (2023) | source tracks (fine-tuned on 3 Bach's chorales) | 1569469 | N/A |
| Hyun et al. (2022) | metadata (BPM=100, Key=A major, Time Signature=4/4, Pitch Range=mid) | 11144 | 13677310 |
| Lu et al. (2023) | text ("Bach's chorale with SATB choir; Moderato; 4/4; A major") | 947659 | ˜120000000 |
| MusicAOG (ours) | hierarchical graph (Structural nodes with attributes same as 3 Bach's chorales) | 3 | 289 |

and the employed sampling technique proficiently generates symbolic music that closely aligns with the data distribution.

Three alternative models were trained to generate MIDIs of music. The controlled schemes among these models vary, but we adhere to the sampling condition capable of describing Bach's music, which served as the dataset for learning and templates for resampling. The detailed generation condition is elucidated in Table 2.

For subjective comparison, we invited 11 musicians possessing at least basic music analysis education. A questionnaire was crafted to score the music across 7 dimensions to scrutinize the model's performance in music composition. Participants were required to assign an integer between 0 and 10 (inclusive) to evaluate each sample music on each dimension. The label names of dimensions and the means and standard deviations of scores received are disclosed in Table 1. Evidently, our model garnered the highest average scores on every dimension, implying superior ability in professional composition.

The objective comparison is not entirely applicable to our model, given its non-reliance on deep learning, and any objective evaluation metrics can serve as the feature descriptors for our model to learn from. Nonetheless, comparisons of objective features of models are furnished in Table 2. A notable advantage of our model is its one-shot nature, negating the need for training on a vast dataset, with a smaller number of parameters to learn. The control conditions of our model can be refined to a finer granularity, imparting certain flexibility in our controllable generation compared to other methods.

Notably, MusicAOG not only encapsulates music scores in western traditions but also accommodates musical notation from diverse cultures. The representability of MusicAOG is explored in the supplementary materials, with example parse graphs for scores in different musical notation.

## 8 CONCLUSION AND FUTURE WORK

This work introduces a novel representation for symbolic music using an And-Or Graph (MusicAOG), enabling a holistic representation of both explicit and implicit musical concepts. Building upon this, we formulated an algorithmic framework for learning symbolic music, tested on a small, simplistic dataset. Due to paper constraints, some model details, especially concerning learning and generating music of various styles and modalities, remain unexplored. However, this groundwork opens several avenues for future research: (1) Extending the attributes and depth of MusicAOG could facilitate representation of more complex music scores and MIDIs. (2) Given that the construction of a music parse graph necessitates a time-intensive labeling process, leveraging the learned energy model as prior knowledge to parse unseen music scores could enrich our dataset. (3) Current manually designed descriptors and proposers in our feature bank and sampling algorithm may not sufficiently capture music composition nuances. Future work could also look into employing neural networks to improve our model's performance, akin to efforts in image synthesis(Lu et al., 2016b).

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
