# OpenReview forum: "MusicAOG: an Energy-Based Model for Learning and Sampling a Hierarchical Representation of Symbolic Music"
_ICLR.cc/2024/Conference — Submitted to ICLR 2024_

### Official Review · Reviewer_T1Mh · 2023-10-24

**Soundness:** 1 poor
**Presentation:** 1 poor
**Contribution:** 2 fair
**Rating:** 3
**Confidence:** 3

**Summary:**

This paper proposes a formalization of music using probabilistic And-or Grammar. Compared to former music analysis model (e.g., GTTM), the proposed method considers more general music data types and formalizes both lower and higher music concepts. The probabilistic model can be learned from data and music is generated via Metropolis-Hastings algorithm.

**Strengths:**

Originality: There is only a few formal methods for music and even fewer that are ready to be applied for automatic music generation. With this regard, the paper makes an attempt propose a solution to this problem. The music grammar has enough complexity to handle some non-trivial music concepts including both local music relations and long-term music structure.

**Weaknesses:**

It is a challenging task to put everything in one research paper including introduction of a formal approach, learning method, inference method and experiments. The approach introduced in this paper is intriguing, but still lacks some crucial material for readers to capture the main idea and the details.

1. The workflow of this study could be clearer? Is the goal of this study music analysis or music generation? Is the generation conditioned on the learned analysis? What music knowledge is assumed and what is not in analysis and generation? What are the major differences compared to existing generation methods, PCFG approach and GTTM?
2. The images in this paper could be clearer. Figure 1 is crucial for readers to understand the defined grammar. However, the defined edge types are not easily recognizable and lacks sufficient explanation about how is the graph achieved. It is even necessary to try to show the graph at multiple levels separately using multiple figures. Figure 2 is not readable.
3. Section 1 and 2 points the drawbacks of existing works. However, the highlight of this work and the workflow of the proposed method is missing.
4. From section 3 to 6, what are the intuition of the proposed method? Try to use intuitive ways to briefly introduce your results and shows the methods at certain level of abstraction. All details can be systematically summarized in the appendix.
5. For section 7, it is not effective to directly compare the methods with deep generative models. The proposed method is almost rule-based (with learnable parameter) and other methods are data-driven. With only three music samples, it is not convincing to say if the model can be generalized to the scenario where there are more data samples and by various composers in different styles. The attributes in Table 1 are not well-introduced and the scores might not be significant. It would be better if the study first present some generation result for analysis and show the learned concept or pattern. Then, evaluate the generative results objectively or subjectively on different learning settings to show generation quality and generalizability on different datasets.

**Questions:**

The questions are related to the weakness section and summarized as follows:

Q1: Could you introduce the workflow of the study in a clearer way?

Q2:  Could you show examples of defined music AOG (e.g., like Figure 1) in a clearer way? What are defined and what are learned?

Q3: Could you provide an example of the learning and generation process of the model if possible?

Q4: Could you explain more on the generalizability of the method on different music corpus?

---

### Official Review · Reviewer_kK8y · 2023-10-28

**Soundness:** 3 good
**Presentation:** 1 poor
**Contribution:** 2 fair
**Rating:** 3
**Confidence:** 2

**Summary:**

This paper proposes MusicAOG, a novel graph-based representation of symbolic music together with a probabilistic model for sampling a musical piece over the graph. The graph representation copes with hierarchical natures of music and leverages structural nodes, textural nodes, and various types of relations (edges) to represent music in a structured manner. The probabilistic model leverages a set of statistical observational constraints and is optimized via the minimax entropy principle. The subjective evaluation demonstrates a superior musicality compared to baseline models with deeper architectures. A sampled sheet music in Bach style also showcases a decent quality.

**Strengths:**

* The hierarchical graph representation of musicAOG is intuitive and gives much insight from a musicology point of view. Specifically, the division of structural and textural functions reflects human perception of music at different cognitive scales. The application of and-or graph also seems a good fit to describe the diverse possibilities of development of a piece of music.

* The energy-based model is lightweight and flexible for few-shot learning. In this paper, a decent sample result is showcased with a training set of 3 annotated Bach chorales.

**Weaknesses:**

* The paper highlights adopting Metropolis-Hastings sampling to enhance fine-grained control of music generation. However, the reviewer questions the extent of this control. The only controllable parameter, $T$, which encourages more varying and creative results, seems insufficient for fine-grained control, and the concrete way of variation seems not controllable.

* MusicAOG is a probabilistic graph model that can learn from a few annotated pieces and then leverage the learned probabilities to sample a new piece. Considering that different styles/genres of music may underlie varied probability distributions, it makes sense to train the MusicAOG model on a very specific style (like Bach chorales in this paper) and fit that style only. Yet, this may also imply a limited potential for a diverse generative capacity. Considering that Bach chorale is a relatively straightforward form of music, it is not surprising that MusicAOG outperforms those baselines with deeper architectures. On the other hand, Music AOG cannot be trained over large and diverse music corpus in a self- or unsupervised manner as those baselines can do. Hence it is questionable if it can cater to more diversified genres and more complex music forms.

**Questions:**

* The metrical tree in Figure 1 is somewhat hard to parse. How do the synchronization nodes relate to the event tree? And what do the colours in the metrical nodes refer to?

* In page 2 Line 2 "$X$ represents the set of vertices", should this be "set of attributes"?

* It is not particularly clear with what the rules $\mathcal{R}=\\{ \gamma\_1, \cdots, \gamma\_k \\}$ refer to and how they function. Are $\gamma_i$ associated with parameters of the multinomial distribution in the probability model? Are their values learned by inferring from the Lagrange multipliers $\lambda\_\alpha$?

* In Section 6.1 "Once all children of a particular node are sampled, a random selection of two distinct child nodes is made, establishing a variation relation $E_{\mathrm{var}}$ between them." What is the purpose of this operation? The reviewer was considering that the model should be able to learn to sample a variation of a previous period/section node by itself.

---

### Official Review · Reviewer_swR4 · 2023-10-31

**Soundness:** 2 fair
**Presentation:** 2 fair
**Contribution:** 3 good
**Rating:** 5
**Confidence:** 3

**Summary:**

This paper presents MusicAOG, a hierarchical graph representation for modeling symbolic music. The authors extensively detail how the MusicAOG and parse graphs are constructed, and demonstrate their method against state of the art generative models for the music generation task.

**Strengths:**

- The authors present an extensive discussion of the proposed method, going in considerable depth as to how the MusicAOG and parse graphs are constructed.
- The relative efficiency of their method (in terms of number of parameters) against current deep learning-based methods is a clear strength of the model.

**Weaknesses:**

- My biggest concern is the lack of generalizability of the present method. As the authors note that extensive human labeling is needed to create the MusicAOG for even one score, it is hard to imagine how usable the present method would be in large scale deployable settings.
- The evaluation section is relatively weak, as the focus on only a handful of example pieces calls into question how extrapolatable the authors' claims are. I think an overabundance of the paper is dedicated to explaining the framework for MusicAOG, which while is useful, could be traded for a more thorough and extensive evaluation.
- The clarity of the overall paper could be improved, as it is relatively hard to follow along the overall flow during the methods section.

**Questions:**

Most concerns were addressed above. While the proposed framework is novel, it is hard to assess its performance or its generalizability from the present work.

---

### Official Review · Reviewer_296y · 2023-11-02

**Soundness:** 2 fair
**Presentation:** 2 fair
**Contribution:** 3 good
**Rating:** 3
**Confidence:** 4

**Summary:**

This paper proposes a generative of model of symbolic music based on many cost (energy) functions. The proposed method is based on the framework of And-Or Graph (AOG), adn the authors extended the AOG to be suitable for symbolic music data.
Although the size of experiment is not large enough, the evaluation results shows that the proposed method outperforms existing methods.

**Strengths:**

The method proposed seems moderately original.
The experiment, albeit the size is limited, shows that the performance is better than existing methods.
The description of this methodology appears to be generally well structured and clear. (However, there are so many elements involved that it does not seem easy to comprehensively and correctly understand the specifics regarding all of these numerous elements. There are too many symbols defined to follow the discussion quickly.)
The authors claim that their proposed method is so comprehensive that it can represent most music in existence, and indeed, the model appears to be that general. It is important to note that the model is not dependent on Western tonal music models, which is an important aspect of symbolic music models.

**Weaknesses:**

- It appears to have fewer references than a typical ICLR empirical paper. For example, I think that there are various studies that should be included as references for GTTM, such as some of parsers (e.g. by Hamanaka) and some of applications, not only the original book. It may not be directly related to the discussion in this paper, but it would be easier to understand the research background if attention is paid to such peripheral discussions.
- The size of experiment is limited. Only three pieces by Bach were examined. I think the number could be increased more. The use of works by more diverse composers other than Bach and Schumann would also make this paper more compelling. Other polyphonic works including some of major composers like Mozart, Beethoven, Bruckner, Schoenberg, etc. and, many polyphonic choral works from the Renaissance periods, and some of academic composers in modern era. There are probably many composers worth considering.
- Although they say it can be used for music from other cultures, the only specific music shown is classical Chinese music, which is still unsatisfactory in quantity.
- It is not clear enough what is essentially different from the existing symbolic music models.

**Questions:**

- The title of this paper calls it an energy-based model, and it is indeed an energy-based model since it uses the Boltzmann/Gibbs distribution. However, calling it "energy-based" simply because various cost functions are placed in the upper right corner of the exponential seems a bit misleading. Of course, logically, it is not wrong, but if I were the author I would not particularly mention such models energy-based, because it is just a common sense.
- The part where the theory of the maximum entropy method is written is just a decorative description of the known theory, and does not seem to be a particularly new discussion. I feel that a more substantive description could be made without being too formal.
- When I encountered the terms "and-node" and "or-node" in the text, I often feel a little confused because they often look like ordinary conjunction words. I think it would be easier to read if they were italicized like *and*-node, *or*-node, or replaced with symbols like $\wedge$-node, $\vee$-node.
- The figures are too small and I have to enlarge them every time to see what is written. I think it would be better to make figures larger.

---

### Meta-Review · Area_Chair_aSxX · 2023-12-07

**Metareview:**

This paper presents a hierarchical graph-based approach to modeling symbolic music.

Reviewers were generally in agreement that the paper is not ready for publication.  In particular, they. noted several weaknesses with the current draft, including missing references, limited experiments, lack of generalizability, and issues with clarity, among others.

The authors did not provide a rebuttal so there was no further discussion amongst the reviewers.  Given that all reviewers advocated for rejecting the paper, that is my recommendation as well.

**Justification For Why Not Higher Score:**

The authors did not provide a rebuttal for the reviews, and all reviewers are in agreement that the paper is not ready for publication.

**Justification For Why Not Lower Score:**

N/A

---

### Decision · Program_Chairs · 2024-01-16

Reject